# Data-Mining Methodology to Improve the Scientific Production Quality in Turkey Meat and Carcass Characterization Studies

**DOI:** 10.3390/ani14142107

**Published:** 2024-07-19

**Authors:** José Ignacio Salgado Pardo, Francisco Javier Navas González, Antonio González Ariza, José Manuel León Jurado, Nuno Carolino, Inês Carolino, Juan Vicente Delgado Bermejo, María Esperanza Camacho Vallejo

**Affiliations:** 1Department of Genetics, Faculty of Veterinary Sciences, University of Córdoba, 14071 Córdoba, Spain; josalgadopardo@outlook.com (J.I.S.P.); fjng87@hotmail.com (F.J.N.G.); juanviagr218@gmail.com (J.V.D.B.); 2Agropecuary Provincial Centre, Diputación de Córdoba, 14071 Córdoba, Spain; jomalejur@yahoo.es; 3Centro de Investigação Vasco da Gama, Escola Universitária Vasco da Gama, 3020-210 Coimbra, Portugal; nuno.carolino@iniav.pt (N.C.); ines.carolino@iniav.pt (I.C.); 4Instituto Nacional de Investigação Agrária e Veterinária, Polo de Inovação da Fonte Boa—Estação Zootécnica Nacional, 2005-424 Santarém, Portugal; 5Centro de Investigação Interdisciplinar em Sanidade Animal, Faculdade de Medicina Veterinária, Universidade de Lisboa, 1300-477 Lisboa, Portugal; 6Laboratório Associado para a Ciência Animal e Veterinária, Faculdade de Medicina Veterinária, Universidade de Lisboa, 1300-477 Lisboa, Portugal; 7Instituto Superior de Agronomia, Universidade de Lisboa, 1349-017 Lisboa, Portugal; 8Andalusian Institute of Agricultural and Fisheries Research and Training (IFAPA), Alameda del Obispo, 14004 Córdoba, Spain; mariae.camacho@juntadeandalucia.es

**Keywords:** data-mining, turkey meat quality, physical traits, chemical profile, publication quality traits, biostatistical tool

## Abstract

**Simple Summary:**

Simple Summary: Currently, research on livestock production suffers from isolation from other disciplines and a generalist nature, which makes publishing in top-tier journals a very difficult task. This situation is even more drastic when it comes to turkey meat research, which is an underdeveloped area that has historically suffered from a lack of resources compared to other species. For this reason, the aim of the present study is to develop a tool that allows researchers to determine which carcass and meat quality traits are related to increased interest by the scientific community and the quality standards of the journals in which studies are published. Variables improving journal standards include carcass dressing traits, muscle fibers properties, pH, colorimetry, some texture and water captivity traits, and chemical composition. Contrarily, carcass or piece yield is not a recommended variable to be performed in studies, as this parameter did not show a clear impact on publication quality. Finally, measures after 72 h are contraindicated since they showed a correlation with poor journal quality standards. Thus, this work can be used as a guideline for designing turkey carcass and meat quality studies, describing parameters to prioritize in order to maximize the impact quality of publication in the scientific community.

**Abstract:**

The present research aims to describe how turkey meat and carcass quality traits define the interest of the scientific community through the quality standards of journals in which studies are published. To this end, an analysis of 92 research documents addressing the study of turkey carcass and meat quality over the last 57 years was performed. Meat and carcass quality attributes were dependent variables and included traits related to carcass dressing, muscle fiber, pH, colorimetry, water-holding capacity, texture, and chemical composition. The independent variables comprised publication quality traits, including journal indexation, database, journal impact factor (JIF), quartile, publication area, and JIF percentage. For each dependent variable, a data-mining chi-squared automatic interaction detection (CHAID) decision tree was developed. Carcass or piece yield was the only variable that did not show an impact on the publication quality. Moreover, color and pH measurements taken at 72 h postmortem showed a negative impact on publication interest. On the other hand, variables including water-retaining attributes, colorimetry, pH, chemical composition, and shear force traits stood out among the quality-enhancing variables due to their low inclusion in papers, while high standards improved power.

## 1. Introduction

Increasing advances in research and how these advances are valued by the scientific community suggest the need to evaluate the determinants of the impact of publications on researchers. Thus, the search for new methodological alternatives or statistical strategies that improve the potential scope of research studies is necessary. However, due to the increase in researchers and the diminishment of research funds availability [1], publishing in a top-tier journal has become dramatically difficult in recent years [2]. Research on livestock carries additional difficulties for publishing in high-impact journals [2] because of the multidisciplinary nature of animal research; for example, including medical studies together with basic biology, welfare, and animal care in a wide spectrum of species [2]. All of these subjects are commonly included in a few unspecific knowledge areas, such as ‘Biology’, ‘Animal Science and Zoology’, or ‘Food Science and Technology’. This multidisciplinary nature tends to cause those areas to be underrepresented when searching the literature [3]. Another cluster in which livestock research is commonly found is ‘Veterinary Science’; however, animal production sciences perform a secondary role behind medical studies in this area [4]. The most specific cluster for livestock research might be ‘Agriculture, Dairy, and Animal Science’. However, this cluster includes plant and environmental research alongside livestock, where dairy cows and pigs are the most popular species [5]. Moreover, the agriculture and animal production fields suffer from a lack of bidirectional interdisciplinary co-citations compared to other fields [6], which implies additional difficulties in reaching a high impact factor [7]. With regard to poultry, research has mainly focused on nutrition, production, environment, and reproduction, and the cluster ‘products, processing, and marketing’ has moved into a secondary role [8]. Furthermore, private research interests have hindered poultry research, aiming to develop intellectual property market tools [9,10].

Research in poultry meat quality began after World War II in industrialized countries to satisfy the increasing demand for animal protein [11]. As a consequence of exhaustive growth selection in broilers [9], together with industrialized practices in transport and at slaughter, carcass defects and meat quality irregularities appeared [11,12]. This previous lack of research investment in meat quality was reported by Kempster [13], who attributed it to the great challenge for the meat industry to reach quality standards at an affordable price. These first meat characterizations included color, pH, texture-related traits, and fat and connective tissue contents. Since then, meat quality research has developed to include new attributes such as animal welfare during slaughter or muscle morphology and its influence on meat properties [11].

Food quality could be considered the confluence of consumers’ needs with the intrinsic and extrinsic attributes of a food product [14]. Intrinsic quality properties include sensory properties, shelf life, chemical and nutritional attributes, and health aspects [14], while extrinsic attributes include traits less related to the physical product, such as animal welfare or environmental aspects and marketing features, such as brands, origin, and packaging [15]. Due to the historic lack of measurability of extrinsic quality attributes, only intrinsic ones are addressed in this study.

Measuring meat’s intrinsic qualities in terms of carcass and meat physical characteristics is expensive and has an economic impact on carcasses due to value depreciation caused by sample collection. This has led to a widespread visual carcass grading and classification model and simple measurements of carcass shape or subcutaneous fat cover [13]. Grading for poultry carcass quality, similar to other livestock species, consists of the evaluation of a normal shape that is fully fleshed, meaty, and free of defects [16]. In the poultry industry, carcass grading is optional, contrary to health inspections in other species’ industries, such as bovines and pigs (which is specified in Regulation (EU) No. 1308/2013). Due to this, there is a lack of uniform criteria when evaluating carcass characteristics among markets [17].

Meat’s intrinsic qualities also involve health, nutritional value, and sensory quality traits [14]. Consumers’ growing demand for fat-free animal protein sources [18], together with ‘functional food’ properties [19], offers a great opportunity for the poultry meat industry. This has led to a great change in poultry meat quality research over the last 20 years [20]. From simple measures such as carcass weight, pH, or water-holding capacity, poultry meat research has developed other sophisticated measures for analyzing muscle fibers [21] or the definition of the amino acid profile of a piece of meat [22]. This comes together with the ‘less but better’ strategy, a market trend promoting a reduction in meat consumption while eating high-quality products [23]. Consumers’ interests are changing in favor of integrity and traceability, eating enjoyment, and ethical production systems, and they are willing to pay the extra costs that these ‘exclusive’ and high-quality products have [24]. Future topics considered within product quality research will tend to include animal performance and its relationship with product quality, sensory analysis, healthier and functional animal food products, and products derived from indigenous breeds [25]. This is becoming a reality in the turkey industry, where several papers have recently been published addressing the effect of meat and carcass quality traits on genotype. A recent study defined the most important quality variables differencing breeds [26,27,28,29] and successfully developed a statistical tool for breed traceability [27]. However, there is no literature addressing the relationship between these meat and carcass quality traits and interest in them in the scientific community.

Therefore, the present article aims to use a data-mining analysis to identify the primary traits to include in turkey carcass and meat quality studies to make them have a higher impact on the scientific community. This would be beneficial for optimizing the economic resources of research institutions, especially for low-income enterprises, such as local breed-related studies or alternative production systems.

## 2. Materials and Methods

### 2.1. Literature Search Strategy and Exclusion Criteria

Data collection was performed as described in previous research [20,30,31,32]. The different research studies were sought using two platforms, (www.google.scholar.es and www.sciencedirect.com, (accessed on 10 June 2024), and the search was last performed in July 2024. The selection of these repositories was based on the fact that they offer tools for data extraction to its analysis, while other browsers, such as www.webofscience.com/wos/woscc/basic-search and www.ncbi.nlm.gov/pubmed/ (accessed on 10 June 2024) do not. This fact prompted their exclusion as information sources, as suggested by Iglesias Pastrana, Navas González, Ciani, Barba Capote, and Delgado Bermejo [31] and González Ariza, Navas González, Arando Arbulu, León Jurado, Delgado Bermejo, and Camacho Vallejo [20].

The literature search was performed using the following keywords: ‘Meat/carcass quality/traits’, followed by ‘turkey’, ‘Meleagris gallopavo’, or any term semantically related [33]. A total of 92 documents were collected for the present study. The bibliography used was published in the English language from 1968 until 2024. The included parameters are shown within their cluster and accompanied by the references of the works from which they were collected in Table 1.

Information about the papers was included in the analysis, considering the country and continent of precedence, year of publication, the publishing journal, and the publication quality traits, such as journal indexation (Yes/No), database (Not indexed, JCR (Journal Citation Reports), SJR (Scimago Journal and Country Rank), or Scopus), journal impact factor (JIF), quartile, publication area, and JIF percentage. Journal indexation (indexed), database, quartile, publication area (area), JIF, and JIF percentage were the independent variables used in the posterior analysis.

A total of 1.210 individual observations were recorded considering different carcass cuts from which they were obtained: carcass remainder, breast, complete leg, thigh, drumstick, wings, head, neck, feet, shank, back, heart, liver, giblets, kidney, lungs, spleen, pancreas, gallbladder, proventriculus, gizzard (full and empty), stomach, complete intestine, small intestine, cecum, abdominal fat, fat pad, ovary, oviduct, feathers, skin, feather plus skin, blood, and waste.

The aforementioned 28 meat and carcass attributes included in Table 1 were considered in the statistical analysis as dependent variables. Thus, the presence or absence (Yes/No) of the aforementioned dependent variables in each study was collected and used in the statistical analysis.

Since the techniques and procedures used to collect the measurements were standardized in the studies, there was no need to record the specific methodologies and techniques used to determine each particular explanatory variable. This decision was made because, even if differences may exist among standardized techniques, these might be negligible, as supported by scientific evidence [121,122]. Thus, the inclusion of each measurement was used as classification criteria to elaborate a data-mining chi-squared automatic interaction detection (CHAID) decision tree according to the quality of the journal in which the study was published.

### 2.2. Data Analysis

#### 2.2.1. Data-Mining CHAID Decision Tree

In order to classify, predict, interpret, and discretely categorize data manipulation, the data-mining CHAID decision tree was employed using the classification tree routine of commercial software (SPSS Version 26.0 for Windows, SPSS, Inc., Chicago, IL, USA). For each dependent variable, comprised of the different carcass or meat quality traits, a decision tree was developed. A root node, branches, and leaf nodes are included using the CHAID-based algorithm decision support tool. Each internal node in the tree was built around a publication quality characteristic (input variables), while the so-called pre-pruning process met a significance split criterion of the chi-square test (*p* < 0.05).

According to Breiman et al. [123], pruning must be carried out in such a way that trees do not have a large number of branches and that branches that significantly contribute to the overall fit are not overlooked. When the computation of a tree exhaustively depicts the significant relationships across the detection of independent variables, those nodes not contributing to the overall prediction are discarded. Additionally, the CHAID method penalizes the model’s complexity. For this purpose, the statistical analyses were developed through the Bonferroni inequality significant level adjustment. Moreover, Breiman’s method resembles forward stepwise regression with a reduction in the number of steps using chi-squared, in opposition to F-to-enter-based tests. Each branch is the representation of a result of the test (in a number of two or more), and each leaf node reflects a category level of the target variable. Thus, decisions are made at each nodal point, and each data record continues down through the tree along a path until the record reaches a terminal node [124].

#### 2.2.2. Data-Mining CHAID Decision Tree Reliability: Cross-Validation

Cross-validation was performed once the model was established in order to ensure that the set of significant predictors properly measured the differences between the prediction errors for a tree. It was applied to a new sample and a training sample. Through the use of the complexity parameter and the cross-validated error, cross-validation of the decision tree was performed to define the accuracy of the model when generalized for unseen data. Ten-fold cross-validation was carried out, keeping each individual observation in either the training sample or study data [125]. The resubstitution error rate is employed to measure the proportion of misclassified original observations by various subsets of the original tree. This is performed to determine the shortest tree with the greatest number of significant relationships. On the other hand, the lowest resubstitution rate is not always the optimal choice because this tree will present a bias. Similarly, large trees will introduce random variation into the predictions by over-fitting outliers. For these reasons, X-fold cross-validation is used to obtain a cross-validated error rate rather than selecting a tree based on the resubstitution error rate. X-fold cross-validation involves the creation of a number of X-random subsets of the original data, setting one portion aside as a test set, constructing a tree for the remaining X-1 portions, and evaluating the tree using the test portion. An estimate of the error is evaluated, and this is repeated for all portions. Adding up the error across the X portions represents the cross-validated error rate. The tree exhibiting the lowest cross-validated error rate is selected due to its good data-fitting properties.

## 3. Results

### 3.1. Study Georeferencing

Figure 1 shows the country distribution of turkey meat and carcass quality studies used in the present research. Most papers included in this research are from the USA (13 studies), India (12 studies), and Canada (10 studies). However, America is the continent that leads the ranking of research study production, followed by Europe, with 32 and 27 studies, respectively. Within Europe, Poland is the country with the most papers (6 studies), followed by Bulgaria (5 studies), and Italy and Germany share the third position, with 4 studies each.

### 3.2. Data-Mining CHAID Decision Tree: Splitting, Pruning, and Building

Appendix A represents the different data-mining CHAID decision trees obtained from the chi-square dissimilarity matrices built in the present study for each dependent variable.

The traits that showed a higher inclusion percentage were carcass/piece yield (71.7%) and slaughter weight (70.5%). However, while slaughter weight showed a moderately positive effect on the journal’s impact factor, carcass and piece yield do not seem to have a positive effect on journal standards.

On the other hand, those traits showing a consistent improver effect on journal standards while being less frequently included were cholesterol (0.6%), cooking loss (0.7%), shear force (4.0%), drip loss (4.6%), pH (5.0%), water-holding capacity (8.2%), ash (10.3%), pH24 (10.7%), b* meat (11.1%), a* meat (11.2%), moisture (11.2%), L* meat (11.4%), protein (12.8%), and fat (12.8%).

### 3.3. Data-Mining CHAID Decision Tree: Splitting, Pruning, and Building

Finally, the robustness and the validity of the obtained results were cross-validated. For this, the number of erroneously classified observations was computed. For each of the trees obtained, the different risk estimates and standard errors computed by applying the cross-validation test did not differ from the results of the model without the cross-validation test. Thus, the stability of the used model was guaranteed. Values for risk estimates and standard errors for methods applied in each tree are shown in Appendix A.

## 4. Discussion

The present study develops an updated evaluation of international research studies focusing on the inclusion criteria of the different traits analyzed in carcass and meat characterization in the turkey species worldwide. In this aspect, North America, Europe, and Asia are the main sources of studies focused on turkey meat and carcass quality. These same regions overlap with those where the majority of papers in high-standard journals are published [6]. South America, despite being the clade of turkey domestication [126], shows a low rate of publication of papers regarding carcass and meat quality. This could be due to a lack of investment in research compared to high-income countries [127], together with the great cost of meat and carcass quality research [13]. In addition to this, special attention to local breeds has been shown in certain countries such as Mexico, Egypt, Iran, Lebanon, Nigeria, Bulgaria, and Turkey. Native genotypes play an important role in developing countries, where they are mainly reared in backyard farming [20]. On the other hand, research studies in high-income countries are mainly focused on commercial hybrid strains. According to the results obtained in the present study, only 5 of 14 papers (35.7%) based on local genotypes have been published in an indexed journal. These results agree with those described by González Ariza, Navas González, Arando Arbulu, León Jurado, Delgado Bermejo, and Camacho Vallejo [20], which analyzed the variability of meat and carcass quality from worldwide native chicken genotypes and evidenced the limited impact of research involving these populations.

Regarding the traits used in this research, despite no consistent trend in the inclusion of the carcass/piece weight variable in studies (41.7%), its inclusion produced a slight improvement in the impact factor of the journals in which studies were published. The importance of this trait lies in the fact that it allows the estimation of the carcass and its components’ yield [68]. On the other hand, the weight of those primary cuts of the carcass, such as breast, thighs, drumsticks, wings, and back, is considered a parameter of growing interest since the acquisition of the whole carcass is no longer popular among consumers, who show a preference for the aforementioned cuts [128]. Moreover, primary cuts adapted to specified market demands of weight and shape have been described as a crucial descriptor of poultry meat sensory features [11]. On the other hand, carcass/piece yield trait is often included in meat quality articles (71.7%). This could be due to the ability to reflect the total amount of meat that is put on the market after the removal of viscera and offal from the carcass [128]. Furthermore, the estimation of the carcass components’ yield is a valuable source of information for producers and intermediaries as long as each primary cut has a different market value [12,71,129]. Hence, knowledge of carcass components’ yield could be applied to each genotype selection program targeting those higher-valued cuts [130]. In addition, carcass dressing is a particularly interesting trait in turkeys, as it is the poultry species with the highest carcass dressing percentages [131]. However, its inclusion did not show a clear effect on the standard of journals, according to the results of the present study. This could suggest that this parameter is a basic carcass attribute included in every kind of study.

Research studies have shown a low tendency to include the cold carcass weight trait in turkey meat quality studies (14.0%), even though its presence is correlated with studies of greater interest for high-impact journals. This could be due to the weight of the cold carcass being widely measured in the meat industry to control evaporative losses during carcass refrigeration [132], which can reach 1–3% of carcass weight in turkeys [90]. However, the slaughter body weight explains 95% of the variation observed in the cold carcass weight trait [101]. This fact might explain the low percentage of inclusion of this variable in carcass characterization studies.

The slaughter weight variable showed a high tendency of inclusion in studies (70.5%) and has a moderate positive correlation to the journal’s impact factor. It is widely used in the turkey industry as a grading factor in modern hybrid strains, distinguishing a heavy, medium-heavy, and medium type [131]. The high use of this variable in the studies allows identifying carcass dressing percentage [68]. Moreover, slaughter weight has also been described to have implications in carcass components’ proportions and dressing percentages [128], as well as intrinsic meat quality traits [11,80].

Even though including muscle fiber diameter has been shown to improve the journals’ interest degree in the studies, muscle fiber characteristics have not been as deeply studied meat quality parameters in poultry as in other livestock species, despite turkey displaying similar meat quality traits [34] and muscle abnormalities as some mammals [17,133]. Furthermore, fiber morphology could be used as a meat quality and animal welfare indicator as soon as it has been correlated to several muscle pathologies [134]. This could cause a change in the journal’s attention to this parameter in turkey meat quality studies, performing as a journal standard improver. On the other hand, muscle growth occurs in turkeys through an increase in the muscle fiber diameter and not by hyperplasia, as in many other species [34]. This fact may explain the reason for the inclusion of muscle density parameters in some studies.

pH measurements are widely spread through meat quality studies. Moreover, the inclusion of pH at slaughter and 24 h postmortem has been reported to significantly improve the journal’s impact factor percentile (JIF %). Meat acidification after slaughter has been described to regulate proteolysis and postmortem muscle contraction, playing an important role in meat tenderness [14,135]. As a major indicator of postmortem biochemical changes, pH has been reported to be correlated to other meat characteristics such as color, drip loss, tenderness, and juiciness [13]. pH is also an indicator of PSE (pale-soft-exudative) meat alterations, which have been described to occur in 40% of turkey breasts [11]. Moreover, early pH measurements have shown a correlation with the technological quality of meat processing in terms of drip loss and product shelf life [80]. On the other hand, pH measurements taken 24 h postmortem are commonly used to draw pH decline curves due to their impact on meat processing [80]. However, pH measurements at 72 h are barely included in papers (0.2%) and show a negative correlation with the journal quartile in which studies are published. This could be because the pH decline in poultry is reached just a few hours postmortem [11,14,136]. Despite this time being influenced by breed, slow-growing turkey breeds have been reported to reach 4 times longer than high-performance strains, in which acidification processes have been described to take only 4 h [136].

The inclusion of color measurements (L*, a*, b* coordinates of CIELAB color space) was strongly positively correlated with the journal’s impact factor. Meat and skin color traits are the first quality attributes noticed by the consumer at the time of purchase, hence their importance [14,128]. Consumers tend to prefer poultry meat that has a color similar to what they are used to, demanding white/pink meat [137] and penalizing extreme paleness and darkness [138]. Furthermore, a growing market niche demands yellowness in poultry meat [139] due to an association with freshness and free-range and natural feed systems by the consumer [140]. Meat color has also been associated with muscle alterations in poultry slaughterhouses. Unlike chicken, heat stress and excitation cause pale meat, similar to those PSE issues in pork, instead of reddish carcasses [17,141]. In addition, meat color is an interesting parameter for meat quality studies because of its relationship with pH, water-holding capacity, cooking loss, and other textural properties [11,136]. Previous authors reported that lighter meat samples (higher L* values) are negatively correlated with water-holding capacity in turkeys [142]. Nevertheless, a* and b* values have not been described to have a significant correlation with any physical meat quality parameter [136], except for the white striping pathology [12]. Contrarily, color measurements (L*, a*, and b*) taken at 72 h postmortem showed a correlation with lower journal quality standards in which studies had been published when including this trait. This could be due to a lack of useful information on turkey meat color from 24 h postmortem, as described by Alvarado and Sams [141]. Thus, color and pH measurements taken at 72 h postmortem are not recommended parameters to include in meat turkey quality studies since they do not offer valuable information nor improve the quality standards of the journal where they are published. Similar findings were described by González Ariza, Navas González, Arando Arbulu, León Jurado, Delgado Bermejo, and Camacho Vallejo [20], who recommended the exclusion of those measurements in fowl meat and carcass quality studies, attaching it to a lack of representativity when prior sampling moments had been considered.

Consumers rank palatability and tenderness as the most important attributes of meat eating qualities. However, these traits are the major source of consumer complaints as well [14]. Complaints about turkey meat toughness during the 1970s stimulated research investments to improve this property of this species [12,136]. Nevertheless, the numerous factors influencing meat tenderness and their interrelations have challenged research in this area, where little progress has been made [14]. In this sense, due to the known role of connective tissue in meat toughness, collagen content and solubility have been the most often studied meat tenderness indicators [143]. However, results in the present research show a tendency not to include collagen analysis in turkey meat quality research studies (0.2%). Moreover, its inclusion has evidenced a slight correlation with the journal’s quality standards in which the studies are published. This could be due to the lower importance of hardness or toughness problems in poultry meat in comparison with beef or pork [34]. Furthermore, intramuscular collagen analysis in turkey breast showed a good level of collagen cross-linking and maturation [12] and has been proven to share characteristics with chicken [129] despite the huge difference in age at slaughter. The shear force has a low rate of inclusion in turkey meat quality studies (4.0%) despite having performed as an interest enhancer for higher journal impact factors. Traditionally, the instrumental measurement of shear force by texture analyzers has been one of the most used methods to determine meat tenderness [144]. It has been employed as an alternative to very time-consuming and costly taste panels [145], which historically have been complemented and correlated with meat chemical composition studies [146]. However, shear force estimation shows limitations, as its correlation to meat tenderness depends largely on the muscle, its genetic basis, and production system [14].

Despite tenderness being the most-valued meat quality attribute, flavor and juiciness are considered increasingly important traits [14]. Juiciness is a harder parameter to quantify and is sometimes combined with tenderness [14]. Additionally, juiciness is assessed from a function involving water-holding capacity and cooking loss [128]. However, when trying to develop models to predict sensory texture attributes based on instrumental measurements, a consensus is not common [147]. The inclusion of water-holding capacity has improved the journal’s impact factor in which studies were published. This is in line with literature describing this attribute as the most significant economic trait in meat [13]. However, since the late 1980s, there has been a notable lack of a simple and cost-effective technique for its implementation in slaughterhouses. That might be why, in recent studies, water-holding capacity has been estimated by assessing other meat quality parameters, such as color and pH [72,136,148]. On the other hand, drip loss and cooking loss have shown a low tendency of inclusion in papers (4.6 and 7.0%, respectively) despite their improving effect on the studies’ JIF percentile. Drip loss has been pointed out as one of the main parameters considered by both producers and consumers to give added value to meat products [149]. Moreover, the drip loss trait has been considered a good determiner of water-holding capacity, being the most affected parameter in PSE meat [150], and keeps correlation to the L* value [149], ultimate pH, and cooking loss [150]. Cooking loss, on its side, has an inverse relationship with water-holding capacity [128] and negatively affects both tenderness and juiciness [151]. Due to the water evaporation while cooking and the dripping of water and fat, the leanness of turkey meat and its often skinless cooking make it an important meat quality parameter to determine [151].

Springiness (0.2%), chewiness (0.2%), gumminess (0.2%), and fragmentation index (0.5%) have scarcely been measured in observations. However, they showed a moderate enhancer effect on the journal quality in which papers were published. The employment of springiness, chewiness, and gumminess in meat quality studies has been controversial. Previous authors have reported that these traits are not well correlated to their descriptive texture [20,152,153] and are imprecise in predicting sensory analysis [154]. However, an investigation has shown their possible role as discriminators of meat quality [147]. Contrarily, the fragmentation index has been described as a useful parameter to provide information about toughness and tenderness when comparing different turkey-rearing systems [153].

Although there is a slight tendency for papers not to include the crude chemical components (moisture, proteins, fats, ash, and cholesterol), their inclusion has shown a correlation with improving the studies’ attractiveness by journals with higher quality criteria. Moisture is a determinant factor in meat quality analysis due to its high influence on meat tenderness, juiciness, firmness, appearance, and economic value [155]. Meat proteins have historically been in the spotlight of research since meat has been historically considered a high-value protein source [14]. Moreover, turkey meat has an especially high protein composition, rounding 28%, against 14–18% in other poultry meat [156]. On the other hand, fat composition has been deeply monitored because of its impact on health [18,19]. However, the selection for leanness in poultry meat has led to a negative impact on meat texture attributes since the collagen content of muscles in turkeys with different growth rates is highly variable. Slow-growing birds show a higher collagen content than fast-growing ones [11]. This may be mainly attributed to the lower cross-sectional area of muscle fibers, leading to a higher content of endomysial collagen relative to muscle volume [157]. Turkey carcasses have a high lean content and relatively low fat content, with muscle inclusion ranging between 2 and 5%, which has made it a desirable source of nutrients for consumers who wish to eat healthy food [22,131,156]. Additionally, turkey meat is characterized by a high mineral content that is necessary for satisfying the normal function of many organs of the human body [158]. Finally, dietary cholesterol has become a special interest from both the scientific and consumer perspectives, in order to determine its impact on health. Consumers associate meat as a source of cholesterol with a high implication for obesity, coronary diseases, and cancer [22]. In this sense, turkey could be a slice of particularly interesting meat because it has the lowest level of cholesterol in comparison with other types of meat [156].

## 5. Conclusions

Conclusively, the present research can be used as a guide for the comprehensive evaluation of literature resources that determine which methodology and meat quality parameters should be used in turkey meat quality studies when aiming to publish papers that are of great interest to the scientific community. There is a general tendency to conduct simple studies, but it has been evidenced that as studies add more quality traits and, therefore, become more complex, they are published in journals with higher research quality values. Results show that North America, Europe, and Asia showed a higher level in the scope of research since these regions have historically made a greater economic effort in the development of research studies based on the nutritional quality of turkey-derived products. Despite being highly recorded and included in observations, carcass/piece yield (%) did not exhibit a clear effect on journals’ standards, while carcass/piece weight (kg), despite not being as widely included, showed a consistent effect on journal quality. The close relationship between cold carcass weight and slaughter weight shows that the first trait has shown a low percentage of inclusion in research studies so far despite being an improver of studies’ interest. Among the lowest-included traits with a strong positive effect on standards were cooking loss, shear force, drip loss, pH (at slaughter and after 24 h), water-holding capacity, meat colorimetry (L*, a*, and b* meat), moisture, protein, fat, ash, and cholesterol. On the other hand, results obtained in the present research suggest that some measurements (pH and color traits) taken at 72 h postmortem can be avoided due to the negative influence on the impact of the research. Hence, this study develops a tool to tailor and improve research efficiency while also maximizing the efficacy of economic funding, which is especially useful for low-income initiatives such as local genotypes research.

## Figures and Tables

**Figure 1 animals-14-02107-f001:**
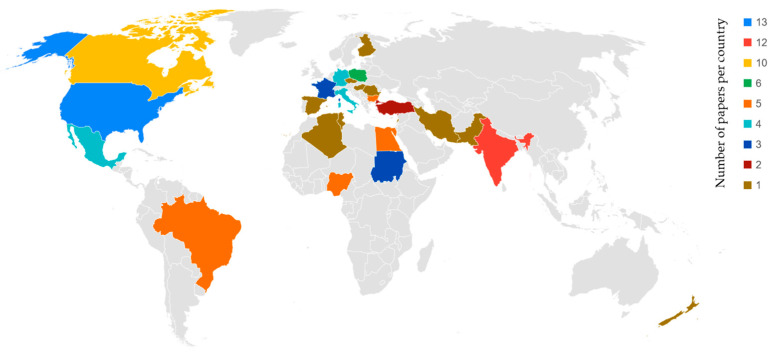
Territorial distribution and number of papers per country.

**Table 1 animals-14-02107-t001:** Cluster of the traits considered in the studies.

Cluster	Trait	
Carcass dressing traits	Carcass/piece weight	[22,28,29,34,35,36,37,38,39,40,41,42,43,44,45,46,47,48,49,50,51,52,53,54,55,56,57,58,59,60,61,62,63,64,65,66,67,68,69,70,71,72,73,74,75,76,77,78,79,80,81,82,83,84,85,86,87,88,89,90,91,92,93,94,95,96,97,98,99,100,101,102,103]
	Carcass/piece yield	
	Cold carcass weight	
	Slaughter weight	
Muscle fiber properties	Muscle fiber diameter	[34]
pH	pH	[22,28,29,34,38,46,47,57,58,59,60,62,70,71,72,73,79,80,82,83,84,85,86,87,88,90,92,95,96,97,98,99,102,103,104,105,106,107,108,109,110,111,112,113,114,115,116,117,118]
	pH 24 h	
	pH 72 h	
Color-related traits	L* meat	[22,28,29,34,37,38,40,46,47,57,58,60,62,70,71,73,79,80,82,86,87,88,90,92,95,96,97,99,102,104,105,108,110,111,115,116,117,118,119,120]
	a* meat	
	b* meat	
	L* meat 72 h	
	a* meat 72 h	
	b* meat 72 h	
Water-holding capacity	Water-holding capacity	[28,29,34,38,40,46,47,48,50,56,57,58,59,60,62,71,73,79,82,83,85,86,87,88,90,92,95,96,98,99,102,103,104,105,106,107,109,110,111,112,113,114,115,117,118,119,120]
	Drip loss	
	Cooking loss	
Texture-related traits	Shear force	[28,29,34,46,56,57,60,71,79,82,83,86,87,90,99,103,105,106,107,110,112,114,115]
	Springiness	
	Gumminess	
	Chewiness	
	Fragmentation index	
Chemical composition	Moisture	[29,34,37,38,45,46,54,56,58,59,60,61,62,65,69,71,79,84,85,86,87,90,92,96,98,102,104,105,106,109,111,112,114,115,118]
	Protein	
	Fat	
	Ash	
	Collagen	
	Cholesterol	

## Data Availability

The data used to support the findings of this study can be made available by the corresponding author upon request.

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
