# Peer review of "Data-Mining Methodology to Improve the Scientific Production Quality in Turkey Meat and Carcass Characterization Studies"

_animals, 2024, doi:10.3390/ani14142107_

Round 1

Reviewer 1 Report

Comments and Suggestions for Authors

In several places in the manuscript, the authors refer to "veterinary research", although I think that meat quality and carcass characterization studies are not exclusively veterinary research, but belong to the biotechnical field of science (animal science (agronomy) and/or food and biotechnological studies).

Given that, based on your research, you came to the conclusion that "Water holding capacity, collagen, springiness, gumminess, chewiness, and frag-48 mentation index were the only variables that did not show an impact on the publication quality", how much is this in collision with by the fact that these are more recent studies and that their impact on publication quality will be visible only later (as opposed to studies that included the determination of slaughterhouse indicators and characteristics of turkey carcasses?  I would like you to look a little more in the discussion at this possibility that could have had an impact on the determined results of your research. Given that the methods and tools used in scientific research are being improved more and more and enable deeper and more diverse research (in this case the quality of meat and the characteristics of turkey carcasses), I suggest that you divide the researched period into several stages (one of the stages should definitely be the last 10-15 years of scientific research) in order to see, for example, the connection between modern turkey carcass traits research and the quality of publications (measured by the usual metric indicators of published articles that you used in this research).

Reviewer 2 Report

Comments and Suggestions for Authors

After reviewing the paper entitled“Data-mining methodology to improve the scientific production quality in turkey meat and carcass characterization studies” the following recommendations are made. Overall, the manuscript is well written; however, there are some important corrections that need to be made before considering publication in animal magazine. The authors did not include 2023 information in the database before analyzing the information and writing the paper. Therefore, if the database is not updated, the manuscript will not be accepted.

Although the revised paper concludes that additional measurements will yield better journals, it is generally agreed.

This manuscript is one figure, from the quality of the manuscript should be added 2-3 figures, while the clarity of the supplementary graphics is not high.

European, North Asian, and Continental Asian contributions to the Turkish case are significant in this paper, but data for 2022,2023, and 2024 are lacking to confirm this conclusion.

L25-26 This statement is not the basis or background of your publication and needs to be rewritten.

L245-247 for each tree obtained, the different risk estimates and standard errors calculated with the cross-validation test were no different from the model results without the cross-validation test. You should add tables or charts to support your results

Round 2

Reviewer 2 Report

Comments and Suggestions for Authors

Accept in current form